# On-Target CRISPR/Cas9 Activity Can Cause Undesigned Large Deletion in Mouse Zygotes

**DOI:** 10.3390/ijms21103604

**Published:** 2020-05-20

**Authors:** Alexey Korablev, Varvara Lukyanchikova, Irina Serova, Nariman Battulin

**Affiliations:** 1Institute of Cytology and Genetics SB RAS, Novosibirsk 630090, Russia; korablevalexeyn@gmail.com (A.K.); taksa_91@mail.ru (V.L.); irina_serova2004@mail.ru (I.S.); 2Laboratory of structural and functional genome organization, Novosibirsk State University, Novosibirsk 630090, Russia

**Keywords:** CRISPR/Cas9, on-target deletions, large deletion, truncation, cytoplasmic microinjections, zygotic microinjections, *Kit* knockout mice

## Abstract

Genome engineering has been tremendously affected by the appearance of the clustered regularly interspaced short palindromic repeats (CRISPR)/CRISPR-associated protein 9 (CRISPR/Cas9)-based approach. Initially discovered as an adaptive immune system for prokaryotes, the method has rapidly evolved over the last decade, overtaking multiple technical challenges and scientific tasks and becoming one of the most effective, reliable, and easy-to-use technologies for precise genomic manipulations. Despite its undoubtable advantages, CRISPR/Cas9 technology cannot ensure absolute accuracy and predictability of genomic editing results. One of the major concerns, especially for clinical applications, is mutations resulting from error-prone repairs of CRISPR/Cas9-induced double-strand DNA breaks. In some cases, such error-prone repairs can cause unpredicted and unplanned large genomic modifications within the CRISPR/Cas9 on-target site. Here we describe the largest, to the best of our knowledge, undesigned on-target deletion with a size of ~293 kb that occurred after the cytoplasmic injection of CRISPR/Cas9 system components into mouse zygotes and speculate about its origin. We suppose that deletion occurred as a result of the truncation of one of the ends of a double-strand break during the repair.

## 1. Introduction

Nowadays, the clustered regularly interspaced short palindromic repeats (CRISPR)/CRISPR-associated protein 9 (CRISPR/Cas9) system, known as the most powerful tool for gene editing, is intensively used in a wide range of biological studies, for conducting various screenings on cell cultures [1,2,3,4] and creating unique genome-modified animal models [5,6,7,8]. To date, two major strategies have been successfully developed and used to create genome-modified animals by using the CRISPR/Cas9 system. The first method, chimera production, consists of combining a recipient morula or blastocyst stage embryo with ES cells with modified genetic background [8,9,10]. The chimeric embryo represents features of both origins, and feather crossings are required to obtain the animal with desired phenotype. The second method, microinjection into mouse zygotes, allows us to modify genomes in different ways, generating knockouts [11], point mutations [12], knock-ins [13,14,15], allele humanization [13], and large chromosome rearrangements [6,16,17,18], as well as producing genetically modified animals, in one step. Both methods have advantages and disadvantages, but microinjection into zygotes is a less-costly and less-time-consuming technique, with a high-efficiency outcome. Additionally, CRISPR/Cas9 technology has given rise to high expectations as a new approach for developing treatment strategies against a spectrum of genetic diseases [19,20] and effective clinical use in vivo and ex vivo. Such a tremendous involvement of the current technique in genomic editing can be explained by the easy application, high efficiency, and remarkable flexibility of this system. However, excessive efficacy of CRISPR/Cas9 system can become a critical point for many genomic editing applications, if an undesigned genome locus will be edited (off-target activity). 

The Cas9 nuclease has an ability to create unspecific cleavages in a non-target locus, specifically if off-target sites are similar in sequence to the desired target sites. An off-target activity may depend on the chromatin state and target site accessibility [21], varies from cell to cell, and needs to be adjusted in a case-by-case manner. To date, multiple off-target prediction tools have been proposed [22,23,24,25], and most of them utilize algorithms based on sequence similarity. Furthermore, numerous molecular biology strategies were described to significantly reduce off-target genome activities: alternative editing enzymes, such as dimeric fusions with FokI [26,27], CpfI [28,29], small Cas9 orthologs [30], CasX [31], and modified spCas9 enzymes [32,33]; improved delivery strategies, such as adeno-associated viral vectors packaging the CRISPR-Cas9 system [34,35], inorganic and lipid nanoparticles [36,37,38,39], and preassembled CRISPR/Cas RNPs [40]; protocol modifications with an intracellular concentration of the nuclease and duration of a nuclease activity [41,42]; truncated gRNAs [43]; and double-nicking strategy [44].

Another type of an incorrect editing outcome is associated with the occurrence of inaccurate genome editing at the target locus [45,46,47,48]. After DNA cleavage and introducing a double-strand break (DSB), cellular systems recognize and repair it. If the DNA repair occurs, using a non-homologous end-joining (NHEJ) path, an error-prone mechanism of the double-strand break repair may happen and result in the appearance of small insertions or deletions (INDELs) near the site of the initial cleavage. Consequently, INDELs can cause frameshift mutations. By applying this molecular mechanism, the production of gene knockouts has been successfully developed [49,50].

At times, unpredicted large genome modifications can take place within the CRISPR/Cas9 editing process. Recently, unplanned deletions at on-target sites were shown to be more than several kilobases [51]. In addition, creating mice with chromosomal rearrangements can be accompanied by hidden complexities, such as a spectrum of structural rearrangements occurring in the same animal [16,52] or rearrangements larger than originally designed [48]. The frequency of these events is extremely low, but the occurrence of them may affect adjacent regulatory elements and cause a serious impact on the activity pattern of surrounding genes. Unanticipated and unpredicted on-target mutations can be considered an even bigger problem of the CRISPR/Cas9 technology than off-target activity, as they do not depend on Cas9 nuclease activities but are associated with the functioning of DNA repair mechanisms in cells. Therefore, to control the phenomenon, it is necessary not to change the CRISPR/Cas9 activity mechanism but to somehow influence the functioning of DSB repair systems.

A set of various high-throughput methods to predict and detect off-target mutations have been developed [53,54,55]. Nonetheless, no method has been proposed for quantifying the entire spectrum of possible mutations that can occur at on-target sites. Large undesigned on-target mutations resulting from the repair of DSB induced by Cas9 are known from a small number of individual examples of such rearrangements [48,51,56]. However, despite the difficulty in detection, this phenomenon should be accurately studied because it can cause dangerous obstacles, especially for clinical applications.

In the current work, we describe the largest, to the best of our knowledge, undesigned on-target deletion that occurred after the injection of CRISPR/Cas9 system components into mouse zygotes.

## 2. Results

### 2.1. Mouse Genome Editing

To examine the role of CCCTC-binding factor (CTCF) protein in maintaining a three-dimensional organization of the genome, we planned to obtain mice through a deletion of CTCF binding sites clustered in the intergenic region between the *Kit* and *Kdr* genes. Based on available data, the particular CTCF cluster of four binding sites, the particular CTCF cluster determines the boundary between two topological domains (Figure 1). To generate the desired genetic modification, we designed two gRNAs targeted upstream and downstream from the selected CTCF sites. The components of CRISPR/Cas9 systems, a mix of two gRNAs and Cas9 mRNA, were delivered into the zygotes through cytoplasmic microinjections. In total, we injected 600 zygotes, where 347 zygotes were transplanted into CD-1 female mice. After all the manipulations, 14 CD-1 female recipients produced 113 live and three dead pups.

### 2.2. Large Unexpected Deletion in the Target Region

Among the 113 mice that were generated, we detected one female pup with an unusual phenotype. A large white spot was identified on the front part of the body (Figure 2A). Genotyping by polymerase chain reaction (PCR) did not reveal the presence of a planned genomic rearrangement in the targeted genomic region in this mouse, as we saw only bands expected for a wild-type allele. Surprisingly, we also did not observe any deletions, inversions, or duplications in the targeted locus. To exclude any accidental failure in the development of the founder, we conducted a crossing of the founder female with a wild-type male. Remarkably, the white spot on the belly was transmitted in accordance with the Mendelian inheritance pattern of dominant alleles (Figure 2B), even though the shape was slightly different. Moreover, we obtained seven F2 offspring from the crossing between two F1 heterozygotes with a white-spotting phenotype. Among the seven animals, we observed two pups with typical phenotypes of anemia and pale skin color. Moreover, the PCR analysis confirmed their homozygosity state. Hence, the current mutation is recessive lethal, as homozygous animals were rarely born and usually died in the perinatal period (Figure 2C).

Several research groups have demonstrated that the white or white-spotting phenotype represents genetic mutations in the locus near and including the *Kit* gene (transmembrane tyrosine kinase receptor) in various organisms, such as alpacas [58], camels [59,60], cats [61], cows [62,63], dogs [64], donkeys [65], goats [66], horses [67], mice [68,69,70], pigs [71], rabbits [72], rats [73], yaks [74], humans [75,76], and even zebrafish [77]. The molecular mechanism behind this phenotype is linked to melanocyte migration and survival maintained by tyrosine-protein kinase KIT receptor. Due to the lack of melanocytes and, as a consequence, of melanin production, these regions remain hypopigmented and represent white spots on the fur.

To establish the molecular nature of the unusual phenotype in our case, the whole genome sequencing of a heterozygous specimen was performed. Remarkably, within the large region surrounding the *Kit* gene (~293 kb), the read coverage was approximately two times lower than that for the rest of the genome (Figure 3), which is the expected evidence of the deletion in one of the alleles, i.e., heterozygous deletion. Based on genomic coordinates determined by next-generation sequencing (NGS) data, we designed primers for deletion borders, amplified those regions, and confirmed the exact coordinates of the deletion, using Sanger sequencing (mm10 chr5:75,588,218-75,881,214; Figure 3). The right border of the deletion occurred at the expected cutting region for gRNA2, three nucleotides away from the PAM sequence, whereas the left border moved ~293 kb toward the *Kit* gene and took place in the first intron of the *Kit* gene. Thus, the observed deletion removed 20 out of the 21 exons of the *Kit* gene, except for the first one. As the first exon encodes only the 5′-UTR sequence and 22 out of the 979 amino acids (MRGARGAWDLLCVLLVLLRGQT) of the signal peptide, the generated deletion can be referred to as the *Kit* gene knockout.

### 2.3. Off-Target or Truncation

One of the possible reasons that may give rise to such an unplanned deletion can be the result of the mistaken off-target Cas9 activity within the left border region. To test this hypothesis, we checked the list of potential off-target sites for both gRNAs used in the experiment. However, none of the predicted off-target sites was located within the deletion coordinates (Appendix A). Therefore, we believe that this scenario could unlikely have happened.

Furthermore, the regular PCR analysis did not reveal any specimen with the deletion, inversion, or either duplication of the genomic region flanked by the gRNA recognition sites A,B). To understand the molecular basis of the on-targeted deletion occurrence, we characterized Cas9 activities in our particular CRISPR/Cas9 experiment. Because double-stranded DNA breaks typically get repaired by NHEJ in eukaryotic cells, CRISPR/Cas9-induced DNA cleavages usually contain small mutations, such as INDELs near the recognition site. In this way, the presence of INDELs can serve as a positive indicator of the Cas9 activity. To verify the Cas9 activity in the experiment, we amplified and sequenced gRNA recognition sites from nine randomly selected founders, after the CRISPR/Cas9 experiment. Although we found mutations near the gRNA2 recognition site in each founder tested, we did not observe any INDELs in the gRNA1 locus (Figure 4). More likely, due to the gRNA degradation during the purification step, gRNA1 was not involved in the CRISPR/Cas9-mediated DNA cleavage. As a result, we did not detect any animal with anticipated structural rearrangements in the CRISPR/Cas9 experiment. However, the CRISPR/Cas9 complex directed by gRNA2 worked effectively, so all founders that were analyzed had INDELs in this area. Thus, we assume that the unplanned large deletion in the white-spotted founder was, more likely, the product of the DSB repair that occurred in the gRNA2 recognition site.

## 3. Discussion

Genome editing directed by the CRISPR/Cas9 system is in widespread use in molecular biology and in therapeutic and clinical trials, making the editing process relatively easy and allowing for the introduction of various genomic modifications into the genome, with high accuracy and precision. On the basis of the CRISPR/Cas9 strategy, many editing tools have been developed, such as nucleases, nickases, and base editors [32,44,78,79]. However, the spCas9 nuclease remains in the highest demand and serves as the most commonly used tool for introducing double-stranded breaks at the desired location in the genome. Using the CRISPR/Cas9 system, researchers can produce the DSB with high efficiency in the target position, but the process of DSB repair is maintained by internal cellular repair systems independent of scientists. Thus, the repair mechanism can occur in different ways and sometimes leads to unpredictable consequences. In this work, we describe the case of an extremely large deletion that happened at the target site after the injection of CRISPR/Cas9 system components into mouse zygotes. To the best of our knowledge, this is the largest deletion that has occurred after the repair of CRISPR/Cas9 introduced DSB in zygotes.

At the present time, there is no technology for an unbiased, high-throughput search for large-scale rearrangements arising from the repair of a DSB. Usually, either PCR or Southern blot is used to detect genomic editing events. PCR serves as a very convenient detection method, but any mutation that destroys a primer annealing site leads to false-negative results [80]. This problem can be partially solved if the primers are moved away from the DSB site as far as possible [81]. However, PCR amplification of the fragments larger than 2 kb is inconvenient for routine use. Therefore, any rearrangement of more than a few kilobases cannot be detected by PCR. In our case, we were not able to detect the presence of deletion in heterozygote animals: Due to removal of a primer annealing site in one homolog, heterozygote was undistinguishable from a wild-type animal (Appendix A). However, when genotyping a homozygous animal, deletions like the one we described here, can be easily detected by a complete lack of amplification due to the loss of primer annealing sites in the both alleles. PCR analysis will not reveal rearrangements with unknown boundaries if genome editing does not involve changing a sequence copy number (such as inversions or translocations). Southern blots can, more likely, detect almost any rearrangement, but this method is time-consuming and inefficient. Moreover, in most cases, one cannot reconstruct the exact nucleotide structure of the rearrangement from the Southern blot picture, which makes this technology not informative enough.

Apparently, third-generation sequencing methods providing long reads may become the most promising approach for unanticipated structural variation detection. Oxford nanopore sequencing technology was successfully used to characterize a complex mutation that occurred during the editing of a mouse zygote genome. The recent work described in Miyamoto et al. (2019) has demonstrated the appearance of an inversion (approximately 5 kb) and deletion of a fragment between two introduced DSBs, using the Oxford Nanopore method [82].

The lack of detection methods makes it almost impossible to accurately assess the occurrence frequency of such large rearrangements. For instance, in the work of Adikusuma et al. (2018), 57 out of 127 animals that were tested had deletions in the target site of more than 100 bp, and the largest deletion detected in the experiment was 2.3 kb in size [81]. At the same time, another group reported that they did not find large deletions within 55 tested embryos after injecting the components of CRISPR/Cas9 into a zygote [83]. The particular features of the experiment design that can cause the occurrence of large unplanned deletions are still unknown. However, it has been shown in experiments on HEK293T cells that the use of a single Cas9^D10A^ nickase for genome editing prevents on- and off-target indels and chromosomal truncations, whereas the Cas9 nuclease approach sometimes leads to unintended rearrangements. Therefore, it can be assumed that DNA double-strand break is the main cause of large on-target deletions. In addition, p53 knockout cells experience significantly more chromosomal truncations at the target site, compared with the control cell population, suggesting a strong involvement of p53 in chromosomal instability induced by CRISPR/Cas9 genome editing [56]. For further progress in this area, the mechanisms of the appearance of large deletions should be further examined. There are still no reliable data on the molecular mechanism of the appearance of large deletions. This is especially important for genome editing in zygotes because several recent studies, including ours, show that major DSB repair pathways may have unusual high activity in early embryos that lead to the formation of concatemers from injected DNA molecules [84,85]. It is possible that such peculiar properties of DSB repair in zygotes will have a role in the formation of large on-target deletion during genome editing.

## 4. Materials and Methods

### 4.1. gRNA Design and gRNA and Cas9 mRNA Preparation

To select gRNAs for the region that determines formation of the TAD boundary in Kit/Kdr locus, we analyzed CTCF Chip-seq data and Hi-C data from Bonev et al. (2017) [57], in HiGlass visualization tool (Figure 1) [86]. We found that a cluster of four CTCF sites forms Kit/Kdr TAD border. Based on their genomic location, we selected regions for CRISPR/Cas9 editing. Two gRNAs were designed by using Benchling software (https://benchling.com). Then, DNA templates for in vitro transcription were amplificated with three oligoDNA-primers (Table 1). The reaction was performed in a total volume of 100 μL containing 1 U of Q5-HF polymerase, 20 μL of ×5 Q5 Reaction Buffer (NEB, M0491), 200 µM of each deoxynucleotide (dATP, dCTP, dGTP, and dTTP), 1000 pmol of T7-gRNA1-FWD or T7-gRNA2-FWD, 100 pmol of gRNA-REV, and 1000 pmol of REV. The reaction was conducted under the following conditions: 11 cycles (20 s at 95 °C, 20 s at 58 °C, and 20 sec at 72 °C) and a final incubation at 72 °C for 2 min. The reaction products were verified on agarose gel and then purified with cleaning columns (Zymo Research, DNA Clean & Concentrator-5). Purified DNA template was measured with NanoDrop 2000 and then proceeded with in vitro transcription.

The in vitro transcription of gRNA oligos was performed with a HiScribe™ T7 High Yield RNA Synthesis Kit (NEB, E2040S, Ipswitch, MA, USA), in accordance with the manufacturer’s protocol. The in vitro transcription of Cas9 mRNA was performed with a HiScribe™ T7 ARCA mRNA Kit (with tailing) (NEB, E2060S, Ipswitch, MA, USA), in accordance with the manufacturer’s protocol. RNA Clean & Concentrator-25 (Zymo Research, R1017, Irvine, CA, USA) was used for gRNAs and Cas9 mRNA purification, in accordance with the manufacturer’s protocol.

### 4.2. Preparation of gRNA and Cas9 mRNA

The in vitro transcription of gRNA oligos was performed with a HiScribe™ T7 High Yield RNA Synthesis Kit (NEB, E2040S, Ipswitch, MA, USA), in accordance with the manufacturer’s protocol. The in vitro transcription of Cas9 mRNA was performed with a HiScribe™ T7 ARCA mRNA Kit (with tailing) (NEB, E2060S, Ipswitch, MA, USA), in accordance with the manufacturer’s protocol. RNA Clean & Concentrator-25 (Zymo Research, R1017, Irvine, CA, USA) was used for gRNAs and Cas9 mRNA purification, in accordance with the manufacturer’s protocol.

### 4.3. Animals

Six-week-old F1 hybrid CBA/J x C57BL/6J female mice were superovulated with 7.5 ME of PMSG (Folligon, Intervet, Boxmeer, Holland) on the first day 1 (4:00 p.m.) and 7.5 ME of hCG (Chorulon, Intervet, Boxmeer, Holland) on the third day (1:00 p.m). Females were crossed with C57BL/6J males, and on the next morning (fourth day), the females were checked for copulation plugs. The individuals with plugs were euthanized by cervical dislocation, then oviducts were excised, and zygote–cumulus mass complexes were flushed out into an M2 medium (M7167, Sigma-Aldrich) by dissecting the ampulla. Cumulus cells were removed by a hyaluronidase (H3506, Sigma-Aldrich) treatment, and the zygotes were rinsed into a fresh M2 medium. Then, the zygotes were transferred into an M16 medium (M7292, Sigma-Aldrich) and cultivated in the medium drops covered with mineral oil (M8410, Sigma-Aldrich) in a cell culture incubator with 5% CO_2_ in the air.

### 4.4. Microinjection

After the incubation, two pronuclear zygotes were placed into a drop of the M2 medium covered with mineral oil and then microinjected into the cytoplasm by mixing two gRNAs (10 ng/uL each) and mRNA Cas9 (50 ng/uL) diluted in nuclease-free water. Next, injected embryos were cultured for a short time (1–2 h), in drops of the M16 medium covered with mineral oil at 37 °C and an atmosphere of 5% CO_2_. Finally, the embryos survived after microinjection were transplanted into the oviducts of the pseudopregnant CD-1 females (0.5 d.p.c.).

The mice were maintained on a 12-hour light/dark cycle, with ad libitum food and water, in a conventional animal facility. All experiments were conducted at the Department of Experimental Animal Genetic Resources, at the Institute of Cytology and Genetics, SB RAS (RFMEFI61914X0005 and FMEFI61914X0010). All the procedures and technical manipulations with animals were in compliance with the European Communities Council Directive of 24 November 1986 (86/609/EEC) and approved by the Bioethical Committee at the Institute of Cytology and Genetics (Permission N45 from 16 November 2018). 

### 4.5. Genotyping

Genomic DNA was isolated from murine tails by placing them in 500 μL of tail lysis buffer containing 100 mM NaCl, 10 mM Tris pH 8.0, 25 mM EDTA, 0.5% sodium dodecyl sulfate, and 0.2 μg/μL proteinase K and then incubating them at 56 °C, until the tissue completely dissolved. Residual fur and bones were removed by centrifugation, and lysates were transferred to the fresh tubes and treated by a standard phenol–chloroform extraction method, followed by ethanol precipitation. Then, ~20 ng (2 μL) of genomic DNA was used for PCR genotyping. The reaction was performed in a total volume of 25 μL containing 1× Taq AS buffer (67 mM Tris-HCl, pH 8.8; 16.6 mM (NH4)2SO4, 0.01% Tween-20) with 1.5 mM MgCl2, 0.2 mM of each deoxynucleotide (dATP, dCTP, dGTP, and dTTP), 0.4 mM of forward and reverse primers (Table 1 and tab1D), and 1 U of Taq polymerase. The reaction was conducted under the following conditions: initial denaturation for 3 min at 95 °C, 35 cycles (30 s at 95 °C, 30 s at 63 °C, and 2 min at 72 °C), and a final incubation at 72 °C for 3 min. PCR products were analyzed by electrophoresis with 2% agarose gels in a Tris–EDTA–acetate buffer and sequenced, using the BigDye^®^ Terminator v3.1 kit (Thermo Fisher Scientific, Waltham, MA, USA), on an Applied Biosystems 3500 Genetic Analyzer, in accordance with the manufacturer’s recommendations.

### 4.6. Whole Genome Sequencing

Genomic DNA was extracted from the tail tissue of a heterozygous F1 mouse with white spot and subjected to paired-end 2× 150 bp Illumina sequencing, generating approximately 400 million reads. Raw reads were analyzed by using FastQC software, to ensure high data quality, and mapped to the mouse genome (mm10) with Bowtie 2 with default parameters. To perform data visualization, alignments were converted to BAM format, sorted and indexed, using SAMtools, and loaded as the IGV browser track.

### 4.7. Data Availability

All sequencing data will be publicly available upon publication, via NCBI Sequencing Read Archive (SRA) under SRX8102546.

## 5. Conclusions

Here we describe the largest, to the best of our knowledge, undesigned on-target deletion with a size of ~293 kb that occurred after the cytoplasmic injection of CRISPR/Cas9 system components into mouse zygotes. We propose that deletion occurred as a result of the truncation of one of the ends of a double-strand break during the repair.

## Figures and Tables

**Figure 1 ijms-21-03604-f001:**
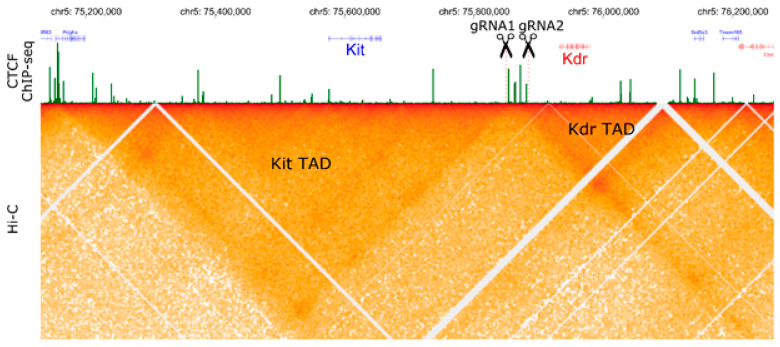
Three-dimensional organization (Hi-C map) and localization of CTCF binding sites (ChIP-seq) for *Kit*/*Kdr* genomic locus in mouse embryonic stem cells, according to [57]. Two gRNAs were selected to delete four CTCF binding sites in the intergenic region between the *Kit* and *Kdr* genes.

**Figure 2 ijms-21-03604-f002:**
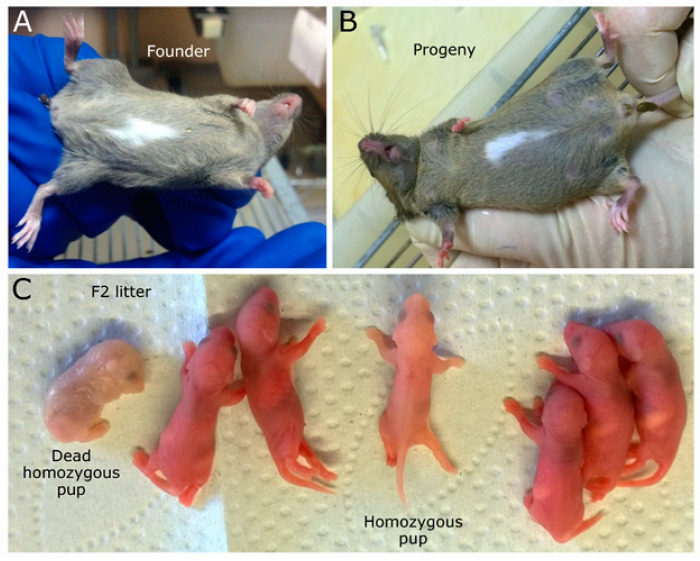
Phenotypic consequence of an on-target deletion. (**A**) Founder with a white spot on the belly. (**B**) F1 progeny with white-spotting phenotype. (**C**) Offspring from the crossing between two heterozygotes with a white spot.

**Figure 3 ijms-21-03604-f003:**
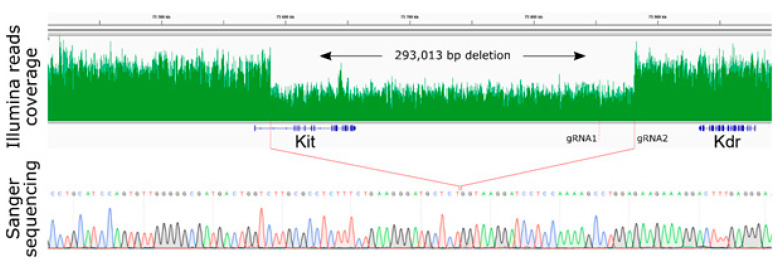
Significant drop in the NGS read coverage visualizes ~293 kb deletion in the heterozygote. Confirmation of the border deletion was performed via Sanger sequencing.

**Figure 4 ijms-21-03604-f004:**
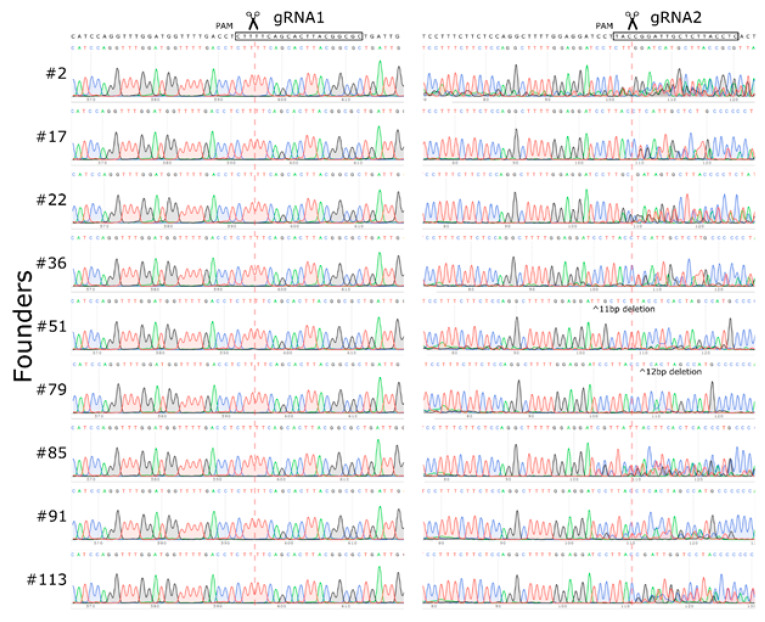
Sanger sequencing for gRNA1 and gRNA2 recognition sites in nine randomly selected founders. Overlapping peaks indicate the presence of heterozygous INDELs, resulting from the repair of DSB.

**Table 1 ijms-21-03604-t001:** DNA oligos for gRNA production and PCR genotyping.

Name	Sequence
T7-gRNA1-FWD	GTTAATACGACTCACTATA*GCGCCGTAAGTGCTGAAAAG*GTTTTAGAGCTAGAAATAGCAAGTTAA
T7-gRNA2-FWD	GTTAATACGACTCACTATA*GAGGTAAGAGCAATCCGGTA*GTTTTAGAGCTAGAAATAGCAAGTTAA
gRNA-REV	AAAAGCACCGACTCGGTGCCACTTTTTCAAGTTGATAACGGACTAGCCTTATTTTAACTTGCTATTTCTAGCTCTA
REV	AAAAGCACCGACTCGGTGCCACTTTTTCAAG
86	TCTTTGTCCTGTTACCGCCC
87	TTGACATGACTCCATGCCCC
88	CCTACGAGCCTTCACGTTGT
89	TGAGGACCGCTGATAGGGAA
90	AAGGCTGTTGTACTGCGTGA
91	ATGATCTCGTGGCGTCATCC
93	GTGCTATGGGAGCCGAAAGA
94	CATAGCCTCTTGCCTTCCGT
179	ATGTTTGGCTGACGCTGAGA

The gRNA recognition sequence is underlined.

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
