# Peer review of "On-Target CRISPR/Cas9 Activity Can Cause Undesigned Large Deletion in Mouse Zygotes"

_ijms, 2020, doi:10.3390/ijms21103604_

Round 1

Reviewer 1 Report

This paper describes a large untargeted deletion cause by CRISPR/Cas9 after cytoplasmic injection of CRISPR editing components into mouse zygote. There are now a number of examples of unintended editing and the use of conventional Cas9 for precise editing is becoming less and less common as researchers are becoming aware of unintended editing of Cas9. Overall, the paper is clearly written and the motivation for the work is clear.

The authors follow up a case of an interesting phenotype in mice and showcase an unexpected deletion of the Kit gene when a region 293kb away is targeted. The authors appropriately use whole genome-sequencing to identify the type of deletion. While the results in the paper are interesting, for it to contribute to wider scientific community the authors should perhaps describe on the method they used to identify the large edits. Mainly the authors note that genotyping did not reveal presence of a planned genomic rearrangement (line 103-104). What did this PCR reveal? If WGS was not performed, how would you have interpreted this data? Would it have been possible to determine that something had gone wrong at this locus without the use of WGS. Additionally, authors should also discuss how the improvements in technology surrounding improved versions of Cas9 may address the problems of conventional Cas9. 

Author Response

Dear Ms. Zhang ,

Thank you for giving me the opportunity to submit a revised draft of our manuscript titled

“On-target CRISPR/Cas9 activity can cause undesigned large deletion in mouse zygotes” to International Journal of Molecular Sciences. We appreciate the time and effort that you and the reviewers have dedicated to providing your valuable feedback on my manuscript. We are grateful to the reviewers for their insightful comments on the manuscript. We have been able to incorporate changes to reflect most of the suggestions provided by the reviewers. We have highlighted the changes within the manuscript in the tracking mode file.

Here is a point-by-point response to the reviewers’ comment.

Sincerely,

Nariman Battulin

Reviewer 1

This paper describes a large untargeted deletion cause by CRISPR/Cas9 after cytoplasmic injection of CRISPR editing components into mouse zygote. There are now a number of examples of unintended editing and the use of conventional Cas9 for precise editing is becoming less and less common as researchers are becoming aware of unintended editing of Cas9. Overall, the paper is clearly written and the motivation for the work is clear.

The authors follow up a case of an interesting phenotype in mice and showcase an unexpected deletion of the Kit gene when a region 293kb away is targeted. The authors appropriately use whole genome-sequencing to identify the type of deletion. While the results in the paper are interesting, for it to contribute to wider scientific community the authors should perhaps describe on the method they used to identify the large edits. Mainly the authors note that genotyping did not reveal presence of a planned genomic rearrangement (line 103-104). What did this PCR reveal?

Reply:

Thank you for this suggestion. In the revised version we specified that as founder animal was heterozygous performed PCR analysis detect only wild type allele specific bands.  This sentence now looks like:

“Genotyping by polymerase chain reaction (PCR) did not reveal the presence of a planned genomic rearrangement in the targeted genomic region in this mouse as we saw only bands expected for a wild type allele.”

If WGS was not performed, how would you have interpreted this data? Would it have been possible to determine that something had gone wrong at this locus without the use of WGS.

Reply:

 We agree with this comment. Therefore, we have added interpretation of PCR analysis in Discussion to emphasize pro and contra of PCR-based detection of unintended rearrangements. This section of Discussion now reads:

“PCR serves as a very convenient detection method, but any mutation that destroys a primer annealing site leads to false-negative results [80]. This problem can be partially solved if the primers are moved away from the DSB site as far as possible [81]. However, PCR amplification of the fragments larger than 2 kb is inconvenient for routine use. Therefore, any rearrangement of more than a few kilobases cannot be detected by PCR. In our case, we were not able to detect the presence of deletion in heterozygote animals: due to removal of a primer annealing site in one homolog, heterozygote was undistinguishable from a wild type animal. However, when genotyping a homozygous animal, deletions like the one we described here, can be easily detected by a complete lack of amplification due to the loss of primer annealing sites in the both alleles. PCR analysis will not reveal rearrangements with unknown boundaries if genome editing does not involve changing a sequence copy number (such as inversions or translocations).”

Additionally, authors should also discuss how the improvements in technology surrounding improved versions of Cas9 may address the problems of conventional Cas9.

Reply:

Thank you for this suggestion. Indeed CRISPR/Cas9 editing tools are very rapidly improved now. However, we did not find any publications in wich new CRISPR techniques would be tested for unintended on-target editing. So it seems hard to fruitfull discuss this aspect now. But thanks to your comment we found that in the original version of the manuscript we missed to mention that genome editing with a single  Cas9 nickase does not cause chromosomal truncation. In the revised version we have added this important point in Discussion.

“However, it has been shown in experiments on HEK293T cells that the use of a single Cas9D10A nickase for genome editing prevents on- and off-target indels and chromosomal truncations whereas Cas9 nuclease approach sometimes leads to unintended rearrangements. Therefore, it can be assumed that DNA double-strand break is the main cause of large on-target deletions. In addition, p53 knockout cells experience significantly more chromosomal truncations at the target site comparing with the control cell population, suggesting a strong involvement of p53 in chromosomal instability induced by CRISPR/Cas9 genome editing [56].”

Reviewer 2 Report

Reicher et al. accidently generated ~293 kb deletion mice in the process of trying to 30 kb deletion of CTCF binding sites clustered in the intergenic region between the Kit and Kdr genes. The authors thought that this super large deletion may be caused by an on-target CRISPR/Cas9 activity. This is a very important report to warn serious side effect in using CRISPR/Cas9 system for medical purpose. There are some minor points the authors need to consider enhancing this interesting manuscript.

Major points:

  1. How did you obtain Fig.1 data? Please describe it in detail.
  2. Page 3, line 113: Fig.2B -> Fig.2C
  3. Please insert all figures with much higher resolution.

Author Response

Dear Ms. Zhang ,

Thank you for giving me the opportunity to submit a revised draft of our manuscript titled

“On-target CRISPR/Cas9 activity can cause undesigned large deletion in mouse zygotes” to International Journal of Molecular Sciences. We appreciate the time and effort that you and the reviewers have dedicated to providing your valuable feedback on my manuscript. We are grateful to the reviewers for their insightful comments on the manuscript. We have been able to incorporate changes to reflect most of the suggestions provided by the reviewers. We have highlighted the changes within the manuscript in the tracking mode file.

Here is a point-by-point response to the reviewers’ comment.

Sincerely,

Nariman Battulin

Reviewer 2

Reicher et al. accidently generated ~293 kb deletion mice in the process of trying to 30 kb deletion of CTCF binding sites clustered in the intergenic region between the Kit and Kdr genes. The authors thought that this super large deletion may be caused by an on-target CRISPR/Cas9 activity. This is a very important report to warn serious side effect in using CRISPR/Cas9 system for medical purpose. There are some minor points the authors need to consider enhancing this interesting manuscript.

Major points:

How did you obtain Fig.1 data? Please describe it in detail.

Reply:

Thank you for pointing this out. We agree with this comment. Therefore, we have added description in Materials and Methods in “gRNA design and gRNA and Cas9 mRNA preparation.”

“To select gRNAs for the region that determines formation of the TAD boundary in Kit/Kdr locus, we analyzed CTCF Chip-seq data and Hi-C data from Bonev et al. (2017) [57] in HiGlass visualization tool (Fig. 1) [86]. We found that a cluster of four CTCF sites forms Kit/Kdr TAD border. Based on their genomic location we selected regions for CRISPR/Cas9 editing.”

Page 3, line 113: Fig.2B -> Fig.2C

Reply:

The suggested correction has been made.

Please insert all figures with much higher resolution.

Reply:

We submitted original figure files with high resolution but I’m afraid that the electronic submission center automatically reduces images resolution during PDF creation.

Round 2

Reviewer 1 Report

As suggested, the authors have now included a discussion on how the interpretation from a single PCR would have looked like. 

This relation to this comment:

"In our case, we were not able to detect the presence of deletion in heterozygote animals: due to removal of a primer annealing site in one homolog, heterozygote was undistinguishable from a wild type animal.'

I would suggest the authors also provide the relevant PCR images for the as a supplementary material. 

Author Response

We would like to thank the reviewers for careful and thorough reading of this manuscript and for the constructive suggestions, which help to improve the quality of this manuscript. Our response follows

Reviewer 1

As suggested, the authors have now included a discussion on how the interpretation from a single PCR would have looked like.

This relation to this comment:

"In our case, we were not able to detect the presence of deletion in heterozygote animals: due to removal of a primer annealing site in one homolog, heterozygote was undistinguishable from a wild type animal.'

I would suggest the authors also provide the relevant PCR images for the as a supplementary material.

Reply:

We agree with the reviewer’s suggestion, in the revised version we have added examples of PCR detection of the rearrangements in Supplementary Figure 1. We also decided to move Figure 5 from the main article to Supplementary to a more comfortable interpretation of PCR images.